# Beyond Convexity: Stochastic Quasi-Convex Optimization

**Elad Hazan**
Princeton University
ehazan@cs.princeton.edu

**Kfir Y. Levy**
Technion
kfiryl@tx.technion.ac.il

**Shai Shalev-Shwartz**
The Hebrew University
shais@cs.huji.ac.il

## Abstract

Stochastic convex optimization is a basic and well studied primitive in machine learning. It is well known that convex and Lipschitz functions can be minimized efficiently using Stochastic Gradient Descent (SGD).

The Normalized Gradient Descent (NGD) algorithm, is an adaptation of Gradient Descent, which updates according to the direction of the gradients, rather than the gradients themselves. In this paper we analyze a stochastic version of NGD and prove its convergence to a *global* minimum for a wider class of functions: we require the functions to be *quasi-convex* and *locally-Lipschitz*. Quasi-convexity broadens the concept of unimodality to multidimensions and allows for certain types of saddle points, which are a known hurdle for first-order optimization methods such as gradient descent. Locally-Lipschitz functions are only required to be Lipschitz in a small region around the optimum. This assumption circumvents gradient explosion, which is another known hurdle for gradient descent variants.

Interestingly, unlike the vanilla SGD algorithm, the stochastic normalized gradient descent algorithm provably requires a minimal minibatch size.

## 1 Introduction

The benefits of using the Stochastic Gradient Descent (SGD) scheme for learning could not be stressed enough. For convex and Lipschitz objectives, SGD is guaranteed to find an $\epsilon$-optimal solution within $O(1/\epsilon^2)$ iterations and requires only an unbiased estimator for the gradient, which is obtained with only one (or a few) data samples. However, when applied to non-convex problems several drawbacks are revealed. In particular, SGD is widely used for deep learning [2], one of the most interesting fields where stochastic non-convex optimization problems arise. Often, the objective in these kind of problems demonstrates two extreme phenomena [3]: on the one hand plateaus—regions with vanishing gradients; and on the other hand cliffs—exceedingly high gradients. As expected, applying SGD to such problems is often reported to yield unsatisfactory results.

In this paper we analyze a stochastic version of the Normalized Gradient Descent (NGD) algorithm, which we denote by SNGD. Each iteration of SNGD is as simple and efficient as SGD, but is much more appropriate for non-convex optimization problems, overcoming some of the pitfalls that SGD may encounter. Particularly, we define a family of *locally-quasi-convex* and *locally-Lipschitz* functions, and prove that SNGD is suitable for optimizing such objectives.

Local-Quasi-convexity is a generalization of unimodal functions to multidimensions, which includes quasi-convex, and convex functions as a subclass. Locally-Quasi-convex functions allow for certain types of plateaus and saddle points which are difficult for SGD and other gradient descent variants. Local-Lipschitzness is a generalization of Lipschitz functions that only assumes Lipschitzness in a small region around the minima, whereas farther away the gradients may be unbounded. Gradient explosion is, thus, another difficulty that is successfully tackled by SNGD and poses difficulties for other stochastic gradient descent variants.

Our contributions:

- We introduce *local-quasi-convexity*, a property that extends quasi-convexity and captures unimodal functions which are not quasi-convex. We prove that NGD finds an $\epsilon$-optimal minimum for such functions within $O(1/\epsilon^2)$ iterations. As a special case, we show that the above rate can be attained for quasi-convex functions that are Lipschitz in an $\Omega(\epsilon)$-region around the optimum (gradients may be *unbounded* outside this region). For objectives that are also smooth in an $\Omega(\sqrt{\epsilon})$-region around the optimum, we prove a faster rate of $O(1/\epsilon)$.

- We introduce a new setup: stochastic optimization of locally-quasi-convex functions; and show that this setup captures Generalized Linear Models (GLM) regression, [14]. For this setup, we devise a stochastic version of NGD (SNGD), and show that it converges within $O(1/\epsilon^2)$ iterations to an $\epsilon$-optimal minimum.

- The above positive result requires that at each iteration of SNGD, the gradient should be estimated using a minibatch of a minimal size. We provide a negative result showing that if the minibatch size is too small then the algorithm might indeed diverge.

- We report experimental results supporting our theoretical guarantees and demonstrate an accelerated convergence attained by SNGD.

## 1.1 Related Work

Quasi-convex optimization problems arise in numerous fields, spanning economics [20, 12], industrial organization [21] , and computer vision [8]. It is well known that quasi-convex optimization tasks can be solved by a series of convex feasibility problems [4]; However, generally solving such feasibility problems may be very costly [6]. There exists a rich literature concerning quasi-convex optimization in the offline case, [17, 22, 9, 18]. A pioneering paper by [15], was the first to suggest an efficient algorithm, namely Normalized Gradient Descent, and prove that this algorithm attains $\epsilon$-optimal solution within $O(1/\epsilon^2)$ iterations given a differentiable quasi-convex objective. This work was later extended by [10], establishing the same rate for upper semi-continuous quasi-convex objectives. In [11] faster rates for quasi-convex optimization are attained, but they assume to know the optimal value of the objective, an assumption that generally does not hold in practice.

Among the deep learning community there have been several attempts to tackle plateaus/gradient-explosion. Ideas spanning gradient-clipping [16], smart initialization [5], and more [13], have shown to improve training in practice. Yet, non of these works provides a theoretical analysis showing better convergence guarantees. To the best of our knowledge, there are no previous results on stochastic versions of NGD, neither results regarding locally-quasi-convex/locally-Lipschitz functions.

## 1.2 Plateaus and Cliffs - Difficulties for GD

Gradient descent with fixed step sizes, including its stochastic variants, is known to perform poorly when the gradients are too small in a plateau area of the function, or alternatively when the other extreme happens: gradient explosions. These two phenomena have been reported in certain types of non-convex optimization, such as training of deep networks.

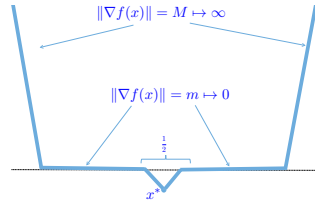

Figure 1: A quasi-convex Locally-Lipschitz function with plateaus and cliffs.

Figure 1 depicts a one-dimensional family of functions for which GD behaves provably poorly. With a large step-size, GD will hit the cliffs and then oscillate between the two boundaries. Alternatively, with a small step size, the low gradients will cause GD to miss the middle valley which has constant size of $1/2$. On the other hand, this exact function is quasi-convex and locally-Lipschitz, and hence the NGD algorithm provably converges to the optimum quickly.

## 2 Definitions and Notations

We use $\|\cdot\|$ to denote the Euclidean norm. $\mathbb{B}_d(\mathbf{x}, r)$ denotes the $d$ dimensional Euclidean ball of radius $r$, centered around $\mathbf{x}$, and $\mathbb{B}_d := \mathbb{B}_d(0, 1)$. $[N]$ denotes the set $\{1, \ldots, N\}$.

For simplicity, throughout the paper we always assume that functions are *differentiable* (but if not stated explicitly, we do not assume any bound on the norm of the gradients).

**Definition 2.1.** *(Local-Lipschitzness and Local-Smoothness) Let $\mathbf{z} \in \mathbb{R}^d$, $G, \epsilon \geq 0$. A function $f : \mathcal{K} \mapsto \mathbb{R}$ is called $(G, \epsilon, \mathbf{z})$-Locally-Lipschitz if for every $\mathbf{x}, \mathbf{y} \in \mathbb{B}_d(\mathbf{z}, \epsilon)$, we have*

$$|f(\mathbf{x}) - f(\mathbf{y})| \leq G\|\mathbf{x} - \mathbf{y}\| \ .$$

*Similarly, the function is $(\beta, \epsilon, \mathbf{z})$-locally-smooth if for every $\mathbf{x}, \mathbf{y} \in \mathbb{B}_d(\mathbf{z}, \epsilon)$ we have,*

$$|f(\mathbf{y}) - f(\mathbf{x}) - \langle \nabla f(\mathbf{y}), \mathbf{x} - \mathbf{y} \rangle| \leq \frac{\beta}{2}\|\mathbf{x} - \mathbf{y}\|^2 \ .$$

Next we define quasi-convex functions:

**Definition 2.2.** *(Quasi-Convexity) We say that a function $f : \mathbb{R}^d \mapsto \mathbb{R}$ is quasi-convex if $\forall \mathbf{x}, \mathbf{y} \in \mathbb{R}^d$, such that $f(\mathbf{y}) \leq f(\mathbf{x})$, it follows that*

$$\langle \nabla f(\mathbf{x}), \mathbf{y} - \mathbf{x} \rangle \leq 0 \ .$$

*We further say that $f$ is strictly-quasi-convex, if it is quasi-convex and its gradients vanish only at the global minima, i.e., $\forall \mathbf{y} : f(\mathbf{y}) > \min_{\mathbf{x} \in \mathbb{R}^d} f(\mathbf{x}) \Rightarrow \|\nabla f(\mathbf{y})\| > 0$.*

Informally, the above characterization states that the (opposite) gradient of a quasi-convex function directs us in a *global* descent direction. Following is an equivalent (more common) definition:

**Definition 2.3.** *(Quasi-Convexity) We say that a function $f : \mathbb{R}^d \mapsto \mathbb{R}$ is quasi-convex if any $\alpha$-sublevel-set of $f$ is convex, i.e., $\forall \alpha \in \mathbb{R}$ the set*

$$\mathcal{L}_\alpha(f) = \{\mathbf{x} : f(\mathbf{x}) \leq \alpha\} \quad \text{is convex.}$$

The equivalence between the above definitions can be found in [4]. During this paper we denote the sublevel-set of $f$ at $\mathbf{x}$ by

$$S_f(\mathbf{x}) = \{\mathbf{y} : f(\mathbf{y}) \leq f(\mathbf{x})\} \ . \tag{1}$$

## 3 Local-Quasi-Convexity

Quasi-convexity does not fully capture the notion of unimodality in several dimension. As an example let $\mathbf{x} = (x_1, x_2) \in [-10, 10]^2$, and consider the function

$$g(\mathbf{x}) = (1 + e^{-x_1})^{-1} + (1 + e^{-x_2})^{-1} \ . \tag{2}$$

It is natural to consider $g$ as unimodal since it acquires no local minima but for the unique global minima at $\mathbf{x}^* = (-10, -10)$. However, $g$ is not quasi-convex: consider the points $\mathbf{x} = (\log 16, -\log 4), \mathbf{y} = (-\log 4, \log 16)$, which belong to the 1.2-sub-level set, their average does not belong to the same sub-level-set since $g(\mathbf{x}/2 + \mathbf{y}/2) = 4/3$.

Quasi-convex functions always enable us to *explore*, meaning that the gradient always directs us in a global descent direction. Intuitively, from an optimization point of view, we only need such a direction whenever we do not *exploit*, i.e., whenever we are not approximately optimal.

In what follows we define local-quasi-convexity, a property that enables us to either *explore/exploit*. This property captures a wider class of unimodal function (such as $g$ above) rather than mere quasi-convexity. Later we justify this definition by showing that it captures Generalized Linear Models (GLM) regression, see [14, 7].

**Definition 3.1.** *(Local-Quasi-Convexity) Let $\mathbf{x}, \mathbf{z} \in \mathbb{R}^d$, $\kappa, \epsilon > 0$. We say that $f : \mathbb{R}^d \mapsto \mathbb{R}$ is $(\epsilon, \kappa, \mathbf{z})$-Strictly-Locally-Quasi-Convex (SLQC) in $\mathbf{x}$, if at least one of the following applies:*

    *1. $f(\mathbf{x}) - f(\mathbf{z}) \leq \epsilon$ .*

2. $\|\nabla f(\mathbf{x})\| > 0$, and for every $\mathbf{y} \in \mathbb{B}(\mathbf{z}, \epsilon/\kappa)$ it holds that $\langle \nabla f(\mathbf{x}), \mathbf{y} - \mathbf{x} \rangle \leq 0$.

Note that if $f$ is $G$-Lispschitz and strictly-quasi-convex function, then $\forall \mathbf{x}, \mathbf{z} \in \mathbb{R}^d$, $\forall \epsilon > 0$, it holds that $f$ is $(\epsilon, G, \mathbf{z})$-SLQC in $\mathbf{x}$. Recalling the function $g$ that appears in Equation (2), then it can be shown that $\forall \epsilon \in (0, 1], \forall \mathbf{x} \in [-10, 10]^2$ then this function is $(\epsilon, 1, \mathbf{x}^*)$-SLQC in $\mathbf{x}$, where $\mathbf{x}^* = (-10, -10)$.

## 3.1 Generalized Linear Models (GLM)

### 3.1.1 The Idealized GLM

In this setup we have a collection of $m$ samples $\{(\mathbf{x}_i, y_i)\}_{i=1}^m \in \mathbb{B}_d \times [0, 1]$, and an activation function $\phi : \mathbb{R} \mapsto \mathbb{R}$. We are guaranteed to have $\mathbf{w}^* \in \mathbb{R}^d$ such that: $y_i = \phi\langle \mathbf{w}^*, \mathbf{x}_i \rangle$, $\forall i \in [m]$ (we denote $\phi\langle \mathbf{w}, \mathbf{x} \rangle := \phi(\langle \mathbf{w}, \mathbf{x} \rangle)$). The performance of a predictor $\mathbf{w} \in \mathbb{R}^d$, is measured by the average square error over all samples.

$$\widehat{\mathrm{err}}_m(\mathbf{w}) = \frac{1}{m} \sum_{i=1}^m \left( y_i - \phi\langle \mathbf{w}, \mathbf{x}_i \rangle \right)^2 . \tag{3}$$

In [7] it is shown that the Perceptron problem with $\gamma$-margin is a private case of GLM regression.

The sigmoid function $\phi(z) = (1 + e^{-z})^{-1}$ is a popular activation function in the field of deep learning. The next lemma states that in the idealized GLM problem with sigmoid activation, then the error function is SLQC (but not quasi-convex). As we will see in Section 4 this implies that Algorithm 1 finds an $\epsilon$-optimal minima of $\widehat{\mathrm{err}}_m(\mathbf{w})$ within poly$(1/\epsilon)$ iterations.

**Lemma 3.1.** *Consider the idealized GLM problem with the sigmoid activation, and assume that* $\|\mathbf{w}^*\| \leq W$. *Then the error function appearing in Equation* (3) *is* $(\epsilon, e^W, \mathbf{w}^*)$-*SLQC in* $\mathbf{w}$, $\forall \epsilon > 0$, $\forall \mathbf{w} \in \mathbb{B}_d(0, W)$ *(But it is not generally quasi-convex).*

### 3.1.2 The Noisy GLM

In the noisy GLM setup (see [14, 7]), we may draw i.i.d. samples $\{(\mathbf{x}_i, y_i)\}_{i=1}^m \in \mathbb{B}_d \times [0, 1]$, from an unknown distribution $\mathcal{D}$. We assume that there exists a predictor $\mathbf{w}^* \in \mathbb{R}^d$ such that $\mathbf{E}_{(\mathbf{x},y)\sim\mathcal{D}}[y|\mathbf{x}] = \phi\langle \mathbf{w}^*, \mathbf{x} \rangle$, where $\phi$ is an activation function. Given $\mathbf{w} \in \mathbb{R}^d$ we define its expected error as follows:

$$\mathcal{E}(\mathbf{w}) = \mathbf{E}_{(\mathbf{x},y)\sim\mathcal{D}}(y - \phi\langle \mathbf{w}, \mathbf{x} \rangle)^2 ,$$

and it can be shown that $\mathbf{w}^*$ is a global minima of $\mathcal{E}$. We are interested in schemes that obtain an $\epsilon$-optimal minima to $\mathcal{E}$, within poly$(1/\epsilon)$ samples and optimization steps. Given $m$ samples from $\mathcal{D}$, their empirical error $\widehat{\mathrm{err}}_m(\mathbf{w})$, is defined as in Equation (3). The following lemma states that in this setup, letting $m = \Omega(1/\epsilon^2)$, then $\widehat{\mathrm{err}}_m$ is SLQC with high probability. This property will enable us to apply Algorithm 2, to obtain an $\epsilon$-optimal minima to $\mathcal{E}$, within poly$(1/\epsilon)$ samples from $\mathcal{D}$, and poly$(1/\epsilon)$ optimization steps.

**Lemma 3.2.** *Let* $\delta, \epsilon \in (0, 1)$. *Consider the noisy GLM problem with the sigmoid activation, and assume that* $\|\mathbf{w}^*\| \leq W$. *Given a fixed point* $\mathbf{w} \in \mathbb{B}(0, W)$, *then w.p.$\geq 1 - \delta$, after* $m \geq \frac{8e^{2W}(W+1)^2}{\epsilon^2} \log(1/\delta)$ *samples, the empirical error function appearing in Equation* (3) *is* $(\epsilon, e^W, w^*)$-*SLQC in* $\mathbf{w}$.

Note that if we had required the SLQC to hold $\forall \mathbf{w} \in \mathbb{B}(0, W)$, then we would need the number of samples to depend on the dimension, $d$, which we would like to avoid. Instead, we require SLQC to hold for a fixed $\mathbf{w}$. This satisfies the conditions of Algorithm 2, enabling us to find an $\epsilon$-optimal solution with a sample complexity that is independent of the dimension.

## 4 NGD for Locally-Quasi-Convex Optimization

Here we present the NGD algorithm, and prove the convergence rate of this algorithm for SLQC objectives. Our analysis is simple, enabling us to extend the convergence rate presented in [15] beyond quasi-convex functions. We then show that quasi-convex and *locally-Lipschitz* objective are SLQC, implying that NGD converges even if the gradients are unbounded outside a small region

---

**Algorithm 1** Normalized Gradient Descent (NGD)

---

**Input**: #Iterations $T$, $\mathbf{x}_1 \in \mathbb{R}^d$, learning rate $\eta$
**for** $t = 1 \ldots T$ **do**
   Update:

$$\mathbf{x}_{t+1} = \mathbf{x}_t - \eta \hat{g}_t \;\; \text{where } g_t = \nabla f(\mathbf{x}_t), \; \hat{g}_t = \frac{g_t}{\|g_t\|}$$

**end for**
**Return:** $\bar{\mathbf{x}}_T = \arg\min_{\{\mathbf{x}_1,\ldots,\mathbf{x}_T\}} f(\mathbf{x}_t)$

---

around the minima. For quasi-convex and *locally-smooth* objectives, we show that NGD attains a faster convergence rate.

NGD is presented in Algorithm 1. NGD is similar to GD, except we normalize the gradients. It is intuitively clear that to obtain robustness to plateaus (where the gradient can be arbitrarily small) and to exploding gradients (where the gradient can be arbitrarily large), one must ignore the size of the gradient. It is more surprising that the information in the direction of the gradient suffices to guarantee convergence.

Following is the main theorem of this section:

**Theorem 4.1.** *Fix $\epsilon > 0$, let $f : \mathbb{R}^d \mapsto \mathbb{R}$, and $\mathbf{x}^* \in \arg\min_{\mathbf{x} \in \mathbb{R}^d} f(\mathbf{x})$. Given that $f$ is $(\epsilon, \kappa, \mathbf{x}^*)$-SLQC in every $\mathbf{x} \in \mathbb{R}^d$. Then running the NGD algorithm with $T \geq \kappa^2 \|\mathbf{x}_1 - \mathbf{x}^*\|^2 / \epsilon^2$, and $\eta = \epsilon/\kappa$, we have that: $f(\bar{\mathbf{x}}_T) - f(\mathbf{x}^*) \leq \epsilon$.*

Theorem 4.1 states that $(\cdot, \cdot, \mathbf{x}^*)$-SLQC functions admit $\text{poly}(1/\epsilon)$ convergence rate using NGD. The intuition behind this lies in Definition 3.1, which asserts that at a point $\mathbf{x}$ either the (opposite) gradient points out a global optimization direction, or we are already $\epsilon$-optimal. Note that the requirement of $(\epsilon, \cdot, \cdot)$-SLQC in any $\mathbf{x}$ is not restrictive, as we have seen in Section 3, there are interesting examples of functions that admit this property $\forall \epsilon \in [0, 1]$, and for any $\mathbf{x}$.

For simplicity we have presented NGD for unconstrained problems. Using projections we can easily extend the algorithm and and its analysis for constrained optimization over convex sets. This will enable to achieve convergence of $O(1/\epsilon^2)$ for the objective presented in Equation (2), and the idealized GLM problem presented in Section 3.1.1. We are now ready to prove Theorem 4.1:

*Proof of Theorem 4.1.* First note that if the gradient of $f$ vanishes at $\mathbf{x}_t$, then by the SLQC assumption we must have that $f(\mathbf{x}_t) - f(\mathbf{x}^*) \leq \epsilon$. Assume next that we perform $T$ iterations and the gradient of $f$ at $\mathbf{x}_t$ never vanishes in these iterations. Consider the update rule of NGD (Algorithm 1), then by standard algebra we get,

$$\|\mathbf{x}_{t+1} - \mathbf{x}^*\|^2 = \|\mathbf{x}_t - \mathbf{x}^*\|^2 - 2\eta\langle \hat{g}_t, \mathbf{x}_t - \mathbf{x}^* \rangle + \eta^2 \;.$$

Assume that $\forall t \in [T]$ we have $f(\mathbf{x}_t) - f(\mathbf{x}^*) > \epsilon$. Take $\mathbf{y} = \mathbf{x}^* + (\epsilon/\kappa)\,\hat{g}_t$, and observe that $\|\mathbf{y} - \mathbf{x}^*\| \leq \epsilon/\kappa$. The $(\epsilon, \kappa, \mathbf{x}^*)$-SLQC assumption implies that $\langle \hat{g}_t, \mathbf{y} - \mathbf{x}_t \rangle \leq 0$, and therefore

$$\langle \hat{g}_t, \mathbf{x}^* + (\epsilon/\kappa)\,\hat{g}_t - \mathbf{x}_t \rangle \leq 0 \;\Rightarrow\; \langle \hat{g}_t, \mathbf{x}_t - \mathbf{x}^* \rangle \geq \epsilon/\kappa \;.$$

Setting $\eta = \epsilon/\kappa$, the above implies,

$$\|\mathbf{x}_{t+1} - \mathbf{x}^*\|^2 \leq \|\mathbf{x}_t - \mathbf{x}^*\|^2 - 2\eta\epsilon/\kappa + \eta^2$$
$$= \|\mathbf{x}_t - \mathbf{x}^*\|^2 - \epsilon^2/\kappa^2 \;.$$

Thus, after $T$ iterations for which $f(\mathbf{x}_t) - f(\mathbf{x}^*) > \epsilon$ we get

$$0 \leq \|\mathbf{x}_{T+1} - \mathbf{x}^*\|^2 \leq \|\mathbf{x}_1 - \mathbf{x}^*\|^2 - T\epsilon^2/\kappa^2 \;,$$

Therefore, we must have $T \leq \kappa^2 \|\mathbf{x}_1 - \mathbf{x}^*\|^2 / \epsilon^2$ . $\qquad\square$

## 4.1 Locally-Lipschitz/Smooth Quasi-Convex Optimization

It can be shown that strict-quasi-convexity and $(G, \epsilon/G, \mathbf{x}^*)$-local-Lipschitzness of $f$ implies that $f$ is $(\epsilon, G, \mathbf{x}^*)$-SLQC $\forall \mathbf{x} \in \mathbb{R}^d$, $\forall \epsilon \geq 0$, and $\mathbf{x}^* \in \arg\min_{\mathbf{x} \in \mathbb{R}^d} f(x)$. Therefore the following is a direct corollary of Theorem 4.1:

**Algorithm 2** Stochastic Normalized Gradient Descent (SNGD)

---

**Input**: #Iterations $T$, $\mathbf{x}_1 \in \mathbb{R}^d$, learning rate $\eta$, minibatch size $b$
**for** $t = 1 \ldots T$ **do**
 Sample: $\{\psi_i\}_{i=1}^b \sim \mathcal{D}^b$, and define,

$$f_t(\mathbf{x}) = \frac{1}{b} \sum_{i=1}^b \psi_i(\mathbf{x})$$

 Update:

$$\mathbf{x}_{t+1} = \mathbf{x}_t - \eta \hat{g}_t \text{ where } g_t = \nabla f_t(\mathbf{x}_t), \; \hat{g}_t = \frac{g_t}{\|g_t\|}$$

**end for**
**Return:** $\bar{\mathbf{x}}_T = \arg\min_{\{\mathbf{x}_1,\ldots,\mathbf{x}_T\}} f_t(\mathbf{x}_t)$

---

**Corollary 4.1.** *Fix $\epsilon > 0$, let $f : \mathbb{R}^d \mapsto \mathbb{R}$, and $\mathbf{x}^* \in \arg\min_{\mathbf{x} \in \mathbb{R}^d} f(\mathbf{x})$. Given that $f$ is strictly quasi-convex and $(G, \epsilon/G, \mathbf{x}^*)$-locally-Lipschitz. Then running the NGD algorithm with $T \geq G^2 \|\mathbf{x}_1 - \mathbf{x}^*\|^2/\epsilon^2$, and $\eta = \epsilon/G$, we have that: $f(\bar{\mathbf{x}}_T) - f(\mathbf{x}^*) \leq \epsilon$.*

In case $f$ is also locally-smooth, we state an even faster rate:

**Theorem 4.2.** *Fix $\epsilon > 0$, let $f : \mathbb{R}^d \mapsto \mathbb{R}$, and $\mathbf{x}^* \in \arg\min_{\mathbf{x} \in \mathbb{R}^d} f(\mathbf{x})$. Given that $f$ is strictly quasi-convex and $(\beta, \sqrt{2\epsilon/\beta}, \mathbf{x}^*)$-locally-smooth. Then running the NGD algorithm with $T \geq \beta \|\mathbf{x}_1 - \mathbf{x}^*\|^2/2\epsilon$, and $\eta = \sqrt{2\epsilon/\beta}$, we have that: $f(\bar{\mathbf{x}}_T) - f(\mathbf{x}^*) \leq \epsilon$.*

**Remark 1.** *The above corollary (resp. theorem) implies that $f$ could have* arbitrarily large gradients and second derivatives *outside $\mathbb{B}(\mathbf{x}^*, \epsilon/G)$ (resp. $\mathbb{B}(\mathbf{x}^*, \sqrt{2\epsilon/\beta})$), yet NGD is still ensured to output an $\epsilon$-optimal point within $G^2\|\mathbf{x}_1 - \mathbf{x}^*\|^2/\epsilon^2$ (resp. $\beta\|\mathbf{x}_1 - \mathbf{x}^*\|^2/2\epsilon$) iterations. We are not familiar with a similar guarantee for GD even in the convex case.*

## 5 SNGD for Stochastic SLQC Optimization

Here we describe the setting of stochastic SLQC optimization. Then we describe our SNGD algorithm which is ensured to yield an $\epsilon$-optimal solution within $\text{poly}(1/\epsilon)$ queries. We also show that the (noisy) GLM problem, described in Section 3.1.2 is an instance of stochastic SLQC optimization, allowing us to provably solve this problem within $\text{poly}(1/\epsilon)$ samples and optimization steps using SNGD.

**The stochastic SLQC optimization Setup:** Consider the problem of minimizing a function $f : \mathbb{R}^d \mapsto \mathbb{R}$, and assume there exists a distribution over functions $\mathcal{D}$, such that:

$$f(\mathbf{x}) := \mathbf{E}_{\psi \sim \mathcal{D}}[\psi(\mathbf{x})] \;.$$

We assume that we may access $f$ by randomly sampling minibatches of size $b$, and querying the gradients of these minibatches. Thus, upon querying a point $\mathbf{x}_t \in \mathbb{R}^d$, a random minibatch $\{\psi_i\}_{i=1}^b \sim \mathcal{D}^b$ is sampled, and we receive $\nabla f_t(\mathbf{x}_t)$, where $f_t(\mathbf{x}) = \frac{1}{b} \sum_{i=1}^b \psi_i(\mathbf{x})$. We make the following assumption regarding the minibatch averages:

**Assumption 5.1.** *Let $T, \epsilon, \delta > 0$, $\mathbf{x}^* \in \arg\min_{\mathbf{x} \in \mathbb{R}^d} f(\mathbf{x})$. There exists $\kappa > 0$, and a function $b_0 : \mathbb{R}^3 \mapsto \mathbb{R}$, that for $b \geq b_0(\epsilon, \delta, T)$ then w.p.$\geq 1 - \delta$ and $\forall t \in [T]$, the minibatch average $f_t(\mathbf{x}) = \frac{1}{b} \sum_{i=1}^b \psi_i(\mathbf{x})$ is $(\epsilon, \kappa, \mathbf{x}^*)$-SLQC in $x_t$. Moreover, we assume $|f_t(\mathbf{x})| \leq M$, $\forall t \in [T], \mathbf{x} \in \mathbb{R}^d$ .*

Note that we assume that $b_0 = \text{poly}(1/\epsilon, \log(T/\delta))$.

**Justification of Assumption 5.1** Noisy GLM regression (see Section 3.1.2), is an interesting instance of stochastic optimization problem where Assumption 5.1 holds. Indeed according to Lemma 3.2, given $\epsilon, \delta, T > 0$, then for $b \geq \Omega(\log(T/\delta)/\epsilon^2)$ samples, the average minibatch function is $(\epsilon, \kappa, \mathbf{x}^*)$-SLQC in $x_t$, $\forall t \in [T]$, w.p.$\geq 1 - \delta$.

Local-quasi-convexity of minibatch averages is a plausible assumption when we optimize an expected sum of quasi-convex functions that share common global minima (or when the different global minima are close by). As seen from the Examples presented in Equation (2), and in Sections 3.1.1, 3.1.2, this sum is generally not quasi-convex, but is more often locally-quasi-convex.

Note that in the general case when the objective is a sum of quasi-convex functions, the number of local minima of such objective may grow *exponentially* with the dimension $d$, see [1]. This might imply that a general setup where each $\psi \sim \mathcal{D}$ is quasi-convex may be generally hard.

## 5.1 Main Results

SNGD is presented in Algorithm 2. SNGD is similar to SGD, except we normalize the gradients. The normalization is crucial in order to take advantage of the SLQC assumption, and in order to overcome the hurdles of plateaus and cliffs. Following is our main theorem:

**Theorem 5.1.** *Fix $\delta, \epsilon, G, M, \kappa > 0$. Suppose we run SNGD with $T \geq \kappa^2 \|\mathbf{x}_1 - \mathbf{x}^*\|^2/\epsilon^2$ iterations, $\eta = \epsilon/\kappa$, and $b \geq \max\{\frac{M^2 \log(4T/\delta)}{2\epsilon^2}, b_0(\epsilon, \delta, T)\}$. Assume that for $b \geq b_0(\epsilon, \delta, T)$ then w.p.$\geq 1-\delta$ and $\forall t \in [T]$, the function $f_t$ defined in the algorithm is $M$-bounded, and is also $(\epsilon, \kappa, \mathbf{x}^*)$-SLQC in $\mathbf{x}_t$. Then, with probability of at least $1 - 2\delta$, we have that $f(\bar{\mathbf{x}}_T) - f(\mathbf{x}^*) \leq 3\epsilon$.*

We prove of Theorem 5.1 at the end of this section.

**Remark 2.** *Since strict-quasi-convexity and $(G, \epsilon/G, \mathbf{x}^*)$-local-Lipschitzness are equivalent to SLQC, the theorem implies that $f$ could have* arbitrarily large gradients *outside $\mathbb{B}(\mathbf{x}^*, \epsilon/G)$, yet SNGD is still ensured to output an $\epsilon$-optimal point within $G^2 \|\mathbf{x}_1 - \mathbf{x}^*\|^2/\epsilon^2$ iterations. We are not familiar with a similar guarantee for SGD even in the convex case.*

**Remark 3.** *Theorem 5.1 requires the minibatch size to be $\Omega(1/\epsilon^2)$. In the context of learning, the number of functions, $n$, corresponds to the number of training examples. By standard sample complexity bounds, $n$ should also be order of $1/\epsilon^2$. Therefore, one may wonder, if the size of the minibatch should be order of $n$. This is not true, since the required training set size is $1/\epsilon^2$ times the VC dimension of the hypothesis class. In many practical cases, the VC dimension is more significant than $1/\epsilon^2$, and therefore $n$ will be much larger than the required minibatch size. The reason our analysis requires a minibatch of size $1/\epsilon^2$, without the VC dimension factor, is because we are just "validating" and not "learning".*

In SGD and for the case of convex functions, even a minibatch of size 1 suffices for guaranteed convergence. In contrast, for SNGD we require a minibatch of size $1/\epsilon^2$. The theorem below shows that the requirement for a large minibatch is not an artifact of our analysis but is truly required.

**Theorem 5.2.** *Let $\epsilon \in (0, 0.1]$; There exists a distribution over convex functions, such that running SNGD with minibatch size of $b = \frac{0.2}{\epsilon}$, with a high probability it never reaches an $\epsilon$-optimal solution*

The gap between the upper bound of $1/\epsilon^2$ and the lower bound of $1/\epsilon$ remains as an open question. We now provide a sketch for the proof of Theorem 5.1:

*Proof of Theorem 5.1.* Theorem 5.1 is a consequence of the following two lemmas. In the first we show that whenever all $f_t$'s are SLQC, there exists some $t$ such that $f_t(\mathbf{x}_t) - f_t(\mathbf{x}^*) \leq \epsilon$. In the second lemma, we show that for a large enough minibatch size $b$, then for any $t \in [T]$ we have $f(\mathbf{x}_t) \leq f_t(\mathbf{x}_t) + \epsilon$, and $f(\mathbf{x}^*) \geq f_t(\mathbf{x}^*) - \epsilon$. Combining these two lemmas we conclude that $f(\bar{\mathbf{x}}_T) - f(\mathbf{x}^*) \leq 3\epsilon$.

**Lemma 5.1.** *Let $\epsilon, \delta > 0$. Suppose we run SNGD for $T \geq \kappa^2 \|\mathbf{x}_1 - \mathbf{x}^*\|^2/\epsilon^2$ iterations, $b \geq b_0(\epsilon, \delta, T)$, and $\eta = \epsilon/\kappa$. Assume that w.p.$\geq 1-\delta$ all $f_t$'s are $(\epsilon, \kappa, \mathbf{x}^*)$-SLQC in $x_t$, whenever $b \geq b_0(\epsilon, \delta, T)$. Then w.p.$\geq 1-\delta$ we must have some $t \in [T]$ for which $f_t(\mathbf{x}_t) - f_t(\mathbf{x}^*) \leq \epsilon$.*

Lemma 5.1 is proved similarly to Theorem 4.1. We omit the proof due to space constraints.

The second Lemma relates $f_t(\mathbf{x}_t) - f_t(\mathbf{x}^*) \leq \epsilon$ to a bound on $f(\mathbf{x}_t) - f(\mathbf{x}^*)$.

**Lemma 5.2.** *Suppose $b \geq \frac{M^2 \log(4T/\delta)}{2} \epsilon^{-2}$ then w.p.$\geq 1-\delta$ and for every $t \in [T]$:*

$$f(\mathbf{x}_t) \leq f_t(\mathbf{x}_t) + \epsilon, \qquad \text{and also,} \qquad f(\mathbf{x}^*) \geq f_t(\mathbf{x}^*) - \epsilon.$$

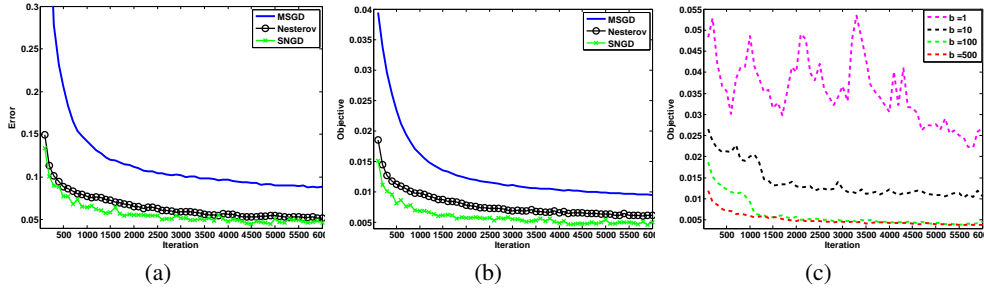

<div align="center">(a)                    (b)                    (c)</div>

Figure 2: Comparison between optimizations schemes. Left: test error. Middle: objective value (on training set). On the Right we compare the objective of SNGD for different minibatch sizes.

Lemma 5.2 is a direct consequence of Hoeffding's bound. Using the definition of $\bar{\mathbf{x}}_T$ (Alg. 2) , together with Lemma 5.2 gives:

$$f(\bar{\mathbf{x}}_T) - f(\mathbf{x}^*) \le f_t(\mathbf{x}_t) - f_t(\mathbf{x}^*) + 2\epsilon, \ \forall t \in [T]$$

Combining the latter with Lemma 5.1, establishes Theorem 5.1. $\qquad\square$

## 6 Experiments

A better understanding of how to train deep neural networks is one of the greatest challenges in current machine learning and optimization. Since learning NN (Neural Network) architectures essentially requires to solve a hard non-convex program, we have decided to focus our empirical study on this type of tasks. As a test case, we train a Neural Network with a single hidden layer of 100 units over the MNIST data set. We use a ReLU activation function, and minimize the square loss. We employ a regularization over weights with a parameter of $\lambda = 5 \cdot 10^{-4}$.

At first we were interested in comparing the performance of SNGD to MSGD (Minibatch Stochastic Gradient Descent), and to a stochastic variant of Nesterov's accelerated gradient method [19], which is considered to be state-of-the-art. For MSGD and Nesterov's method we used a step size rule of the form $\eta_t = \eta_0 (1 + \gamma t)^{-3/4}$, with $\eta_0 = 0.01$ and $\gamma = 10^{-4}$. For SNGD we used the constant step size of $0.1$. In Nesterov's method we used a momentum of $0.95$. The comparison appears in Figures 2(a),2(b). As expected, MSGD converges relatively slowly. Conversely, the performance of SNGD is comparable with Nesterov's method. All methods employed a minibatch size of 100.

Later, we were interested in examining the effect of minibatch size on the performance of SNGD. We employed SNGD with different minibatch sizes. As seen in Figure 2(c), the performance improves significantly with the increase of minibatch size.

## 7 Discussion

We have presented the first provable gradient-based algorithm for stochastic quasi-convex optimization. This is a first attempt at generalizing the well-developed machinery of stochastic convex optimization to the challenging non-convex problems facing machine learning, and better characterizing the border between NP-hard non-convex optimization and tractable cases such as the ones studied herein.

Amongst the numerous challenging questions that remain, we note that there is a gap between the upper and lower bound of the minibatch size sufficient for SNGD to provably converge.

### Acknowledgments

The research leading to these results has received funding from the European Union's Seventh Framework Programme (FP7/2007-2013) under grant agreement n° 336078 – ERC-SUBLRN. Shai S-Shwartz is supported by ISF n° 1673/14 and by Intel's ICRI-CI.

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
