[Reviews · NeurIPS 2015]

Submitted by Assigned_Reviewer_1

This paper looks at the problem of optimizing stochastic local, quasi-convex functions. This encompasses a bigger set of functions than convex functions and the authors provide an example of a function related to training deep networks which falls on the set of local quasi-convex without being convex. They devise algorithms that converges with high probability $\epsilon$ close to the true minima in $O(1/\epsilon^2)$ time for SLQC functions and in $O(1/\epsilon)$ time if the function also has local smoothness properties using an algorithm that is based on normalized gradient descent operated on a minibatch at every iteration.

The paper is well motivated and the mathematics looks clean and easy to follow. The authors also give examples of GLM and noisy GLM which fall under the domain of SLQC functions and hence their algorithm can be used to optimize these classes of objectives as well. There does not seem to exist much previous work on optimizing local quasi-convex objectives and these class of functions is well motivated.

It would be good to see some more experiments for other objectives related to neural network objectives as well as any comparison of the performance of the algorithm with the Isotron algorithm. Finally, the analogy given about gradient descent jumping between very low and very large gradients using SGD is very nice. However it would be good to see a mathematical example related to this. In particular, it would be good to have an example of a function which is SLQC and would have very large gradient values (preventing SGD from converging if we ran on it directly).
Summary: This paper looks at the problem of optimizing stochastic local, quasi-convex functions. This encompasses a bigger set of functions than convex functions and the authors provide an example of a function related to training deep networks which falls on the set of local quasi-convex without being convex. They devise algorithms that converges with high probability $\epsilon$ close to the true minima in $O(1/\epsilon^2)$ time for SLQC functions and in $O(1/\epsilon)$ time if the function also has local smoothness properties using an algorithm that is based on normalized gradient descent operated on a minibatch at every iteration.

Submitted by Assigned_Reviewer_2

Summary: This paper analyzes the stochastic version of normalized version of normalized gradient descent (NGD), which is the first effort to explore the efficacy and property of stochastic normalized gradient descent (SNGD). In order to verify the benefits of NGD in training non-convex optimization problems, this paper introduces a new property, local-quasi-convexity, to prove its convergence to a global minimum. Particularly, they prove that NGD finds an \epsilon-optimal minimum for local quasi convex functions within O(1/ \epsilon^2) iterations. In addition, this paper introduces a new setup: stochastic optimization of locally-quasi convex functions, in which the gradient is estimated using a minibatch of examples. Empirically, this paper reports experimental results by training deep neural networks by comparing with the-state-of-the-arts methods, minibatch SGD and Nesterov's accelerated gradient method.

Quality: This paper is technically sound without obvious mistakes. The theoretical analyses are easy to follow and achieve the expected outcome. The empirical study shows the improvement of the proposed algorithm than the existing methods. This is the first effort for the provable gradient-based algorithm for stochastic quasi-convex optimization. Nevertheless, its training model is relatively small network, that is, a single hidden layer of 100 units. This setup is very likely to be too easy to show the advantage of the algorithm, not to mention the real application of training deep neural network on large datasets.

Clarity:

This paper is easy to follow and well structured. Several typos do exist, however. For example, line 37: 'SGD is guaranteed find' should be 'SGD is guaranteed to find'; line 134: 'Inforamlly' should be 'Informally'; line 185: 'its is' should be 'it is'; line 250: 'and and' should be 'and', etc. The draft should be better checked if accepted.

Originality: In keeping with NIPS objective of rewarding new unexplored areas and research directions (more than incremental advancement of the state of the art), this paper studies an important problem for non-convex optimization and tries the first effort on provable results, which should be scored a high score for originality. It is of significance to move the focus from stochastic convex optimization to stochastic non-convex optimization.

Significance: This work studies an important research problem and is in a good research direction since stochastic non-convex optimization has shown its importance for several years and needs to be solved with solid theoretical guarantee. They define the property, local-quasi-convexity, to extend quasi-convexity and capture the unimodal functions which are not quasi-convex, and then leads to the proof the convergence to the global optimal. However, the empirical studies shown in the paper are not comprehensive or convincing to show that the proposed algorithm could work for training deep neural network since only a single layer model is presented.

Other comments:

The key strength of this paper is to introduce the property, local-quasi-convexity, to extend quasi-convexity and capture the unimodal functions that are not quasi-convex. They propose solutions for locally-quasi-convex optimization with NGD and strictly-locally-quasi-convex optimization with stochastic NGD with both theoretical proofs.

The main concern about this paper lies in the empirical studies. I believe that the proposed algorithm aims to solve the problem of training deep neural network finally. However, in this paper, the model for training is too simple, which is just a single layer with 100 hidden units. Therefore, whether the proposed algorithm could really work for training deep neural network in practice beating SGD is yet to be verified.
Summary: The research direction of the paper is good, and it represents the first effort of generalizing the well-studied stochastic convex optimization to the challenging non-convex optimization problems facing machine learning community. Although the empirical study is not very mature, it represents a starting effort.

Submitted by Assigned_Reviewer_3

The paper introduces a stochastic version of the normalized gradient descent (NGD) algorithm. The author(s) also introduces the concept of strictly locally-quasi-convex (SLQC) functions, and shows that both the noiseless and noisy generalized linear models satisfy the SLQC requirements (although these proofs are deferred to appendix B and C, respectively). The author(s) then proves convergence rates for both the NGD and SNGD algorithms on SLQC functions. Finally, on a very simple neural network architecture (MLP with 100 hidden units, trained on MNIST), the proposed SNGD algorithm is compared to minibatch SGD and SGD with momentum, which are now standard optimization approaches for neural networks. Both the test and training errors with SNGD match (if not beat; though no error bars or repetition numbers are given) that of the momentum approach.

Minor points: - Obviously, the manuscript is slightly over the page limit. - Missing periods: lines 192, 215, 377. - Abstract: the second half of the abstract doesn't read like an abstract; consider boasting the proven convergence rate and the synthetic experiment. - Introduction:

o line 37: "is guaranteed [to] find"

o line 102: "phenomenon" -> "phenomena" - Section 2: slightly inconsistent notation for the ball, specifically with regards to the subscript d. - Section 3:

o line 204: broken sentence

o line 208: "sssume" -> "assume" - Section 5:

o line 318: strictly speaking, the minibatch constituents psi_i(x) could use an index t, but this might clutter notation unnecessarily.

o line 329: "a common global minima": either remove the leading "a" or replace "minima" with "minimum".

o line 343 and 364: strange use of "then". - Discussion: "hereby" -> "herein" - References: needs fixing, e.g., [1] includes "et al."; [11] formatting is inconsistent with the rest. - Appendices: title of H is wrong.
Summary: The paper is clear and well written. The algorithm is well motivated with accompanying theoretical results to guarantee convergence, as well as a simple proof-of-concept experiment on a real experiment (training an MLP on MNIST).

Author Feedback
Author rebuttal: We thank the reviewers for their time and effort, and appreciate their useful comments. We will address all of them in the final version of the paper.

Reviewer 1:
===========
*Experimental part: we presented small scale experiments as a proof-of-concept demonstrating the applicability of the NGD algorithm for non-convex optimization tasks to validate our theoretical findings. We consider this as a first step in examining the performance of NGD in tasks of larger scale.

Reviewer 3:
===========
*"The proposed algorithm is not new in the sense that most practitioners would use mini-batch SGD and make sure to normalize the data and/or solve a constraint optimization variant using a simple projection. This might prevent some of the pathological cases to cause trouble."

-Normalizing the data and using SGD is not equivalent to using NGD (which normalizes the gradients). For example, consider the idealized GLM problem with sigmoid activation (depicted in section 3.1.1), even if we normalize the training data to lie on the unit sphere, plateaus may still occur.

Reviewer 5:
===========
*"It would be good to see some more experiments for other objectives related to neural network objectives as well as any comparison of the performance of the algorithm with the Isotron algorithm"

-It does seem interesting to compare the performance of NGD to the Isotron, and examine the performance on larger scale networks. We certainly intend to pursue this direction in future work.

*"Finally, the analogy given about gradient descent jumping between very low and very large gradients using SGD is very nice. However it would be good to see a mathematical example related to this. In particular, it would be good to have an example of a function which is SLQC and would have very large gradient values"

-Thank you for the suggestion. We will consider adding such an example to section 1.2 in the final version of the paper.